



**Spatio-temporal changes in cryoconite community, isotopic, and elemental composition**
**over the ablation season of an alpine glacier**
**Tereza Novotná Jaroměřská[1], Roberto Ambrosini[2], Dorota Richter[3], Miroslawa**
**Pietryka[3], Przemyslaw Niedzielski[4], Juliana Souza-Kasprzyk[4], Piotr Klimaszyk[5], Andrea**
**Franzetti[6], Francesca Pittino[7], Lenka Vondrovicová[8], Tyler Kohler[1], and Krzysztof**
**Zawierucha[9]\***
*[1]Department of Ecology, Faculty of Science, Charles University, Praha, Czech Republic*
*[2]Department of Environmental Science and Policy, University of Milan, Milan, Italy*
*[3]Department of Botany and Plant Ecology, Wrocław University of Environmental and Life*
*Science, Wrocław, Poland*
*[4]Department of Analytical Chemistry, Faculty of Chemistry, Adam Mickiewicz University,*
*Poznań, Poland*
*[5]Department of Water Protection, Faculty of Biology, Adam Mickiewicz University, Poznań,*
*Poland*
*[6]Department of Earth and Environmental Sciences, University of Milano-Bicocca, Milan, Italy*
*[7]Biodivers. Conserv.Biology, Swiss Federal Research Institute WSL, Birmensdorf, Switzerland*
*[8]Institute of Geochemistry, Mineralogy and Mineral Resources, Faculty of Science, Charles*
*University, Prague, Czech Republic*
*[9]Department of Animal Taxonomy and Ecology, Faculty of Biology, Adam Mickiewicz*
*University, Poznań, Poland*
\* Correspondence: Krzysztof Zawierucha (k.p.zawierucha@gmail.com)



**Abstract:** Cryoconite holes (water-filled reservoirs) on glacier surfaces are important biodiversity hotspots and biogeochemical factories within terrestrial cryosphere. In this study, we collected cryoconite from the ablation zone of the Forni Glacier (Central Italian Alps) over the whole ablation season. We aimed to describe spatial and temporal patterns in: (i) biomass and community structure of photoautotrophs (cyanobacteria, diatoms, and eukaryotic green algae) and invertebrates; (ii) carbon and nitrogen stable isotopic composition of invertebrates and their potential food; and (iii) the organic matter content and general elemental composition of cryoconite. Structure and biomass of cryoconite biota showed spatio-temporal changes over the season. Dominant cyanobacteria were Oscillatoriaceae and Leptolyngbyaceae, while dominant eukaryotic green algae were Mesotaeniaceae and Chlorellaceae. Eukaryotic green algae dominated in the upper part of the ablation zone, while a seasonal shift from algae- to cyanobacteria-dominated communities was observed in the lower part. Some taxa of photoautotrophs appeared only during specific sampling days. Dominant grazers were tardigrades (*Cryobiotus klebelsbergi*). The biomass of tardigrades in the upper part was significantly related to the biomass of eukaryotic green algae indicating that algal communities are likely controlled by grazing. The $\delta^{13}C$ of tardigrades followed fluctuations of $\delta^{13}C$ in organic matter. We did not observe spatial and temporal changes in the general elemental composition of cryoconite. Thus, changes in community structure and biomass are likely dependent on the interplay between phenology, stochastic events (e.g. rainfall), top-down, or bottom-up controls. Our study shows that understanding the ecology of biota in cryoconite holes requires a spatially explicit and seasonal approach.

**Keywords:** top-down control, Forni Glacier, Tardigrada, stable isotopes, phenology, supraglacial habitats



## 1. Introduction

Studies on changes in the distribution, structure, and biomass of organisms in space and time are important for understanding the phenology and resources use of species and their responses to environmental shifts (Post and Stenseth, 1999; Sommers et al., 2019; Vecchi et al., 2021; Walther et al., 2002; Winkel et al., 2022). Glaciers and ice sheets are one of the fastest-changing biomes on Earth (Anesio and Laybourn-Parry, 2012; Zemp et al., 2006). The biological activity on the glacier surface (supraglacial) can affect the surface albedo (reflection of solar radiation) with potential implications to glacier melt dynamic (e.g. Stibal et al., 2012a; Yallop et al., 2012). Understanding the controls on the biodiversity, and phenology of glacial biota is therefore crucial for modelling how climate changes may alter glacial ecosystems (Anesio and Laybourn-Parry, 2012; Gobbi et al., 2021; Stibal et al., 2012a).

Most biological processes within the supraglacial environment occur during the ablation season when air temperature rises, day length is extended, and snow melts (Cameron et al., 2012; Senese et al., 2012; Stibal et al., 2012a). During that time, glacier surfaces provide liquid water and suitable conditions for the activity of myriad organisms from bacteria to invertebrates (e.g. Cameron et al., 2012; Shain et al., 2021; Zawierucha et al., 2018, 2020). On glaciers, the highest biodiversity is found in cryoconite holes, which are small water-filled depressions on the glacier surface formed by a dark sediment (the cryoconite) that lowers the albedo of the glacier surface and melts into the ice (Cameron et al., 2012; Takeuchi et al., 2001a; Wharton et al., 1985). Due to their pond-like structure, cryoconite holes harbour a unique community of organisms from microbes to minute invertebrates (Edwards et al., 2013; Franzetti et al., 2017; Poniecka et al., 2020; Zawierucha et al., 2019a).

Although biological communities on glaciers have been intensively studied over recent years, changes in the community structure of cryoconite over the ablation season have been minimally investigated (e.g. Musilova et al., 2015; Pittino et al., 2018; Takeuchi 2013; Winkel





et al., 2021), and some of these studies have brought contrasting results. For example, Pittino
et al. (2018) showed that the supraglacial microbial community structure in the Alps changed
over the ablation season from a dominance of cyanobacteria to heterotrophic bacteria.
Conversely, on an Arctic glacier, Musilova et al. (2015) showed that community structure
appeared to be stable over the season. Moreover, studies considering the biomass of different
taxa from cryoconite holes are scarce and do not include seasonal observations (e.g. Buda et
al., 2020) although biomass estimation is critical from a mass-balance perspective of
biogeochemistry.
It is known that microinvertebrates can play an important role as consumers in polar
ecosystems (Almela et al., 2019; Shaw et al., 2018; Velázquez et al., 2017). However, their role
in the supraglacial trophic network remains unclear (Novotná Jaroměřská et al., 2021;
Zawierucha et al., 2018), and our understanding of the interactions between microbial
communities and their consumers on glaciers is scant. Even though some studies have
investigated the ecology and community structure of biota in cryoconite together with the
potential food of tardigrades by various methods (Vonnahme et al., 2016; Zawierucha et al.,
2022), evidence related to their potential ecological and trophic roles, including possible top-
down control of the cryoconite ecosystem, remains limited.
Until now, seasonal patterns in the community structure and biomass of both
photoautotrophs and consumers in cryoconite on alpine glaciers have never been studied. Such
missing information on their seasonal evolution prevents the estimation and understanding of
(i) biological diversity on glaciers, since some taxa may appear only in a particular period
during the ablation season (Pittino et al., 2018); (ii) trophic links, which are difficult to resolve
in snapshot studies (e.g. Zawierucha et al., 2018); and (iii) the blooming of photoautotrophs in
cryoconite and the consequent increase of biological activity which may spur supraglacial melt
(Williamson et al., 2020). Moreover, seasonal differences in the community structure of


cryoconite organisms, especially in the production and availability of organic matter (OM), may
affect the composition and nutritional content of glacier meltwater, which is one of the major
sources of water to the proglacial (in front of the glacier) areas and downstream systems in
alpine and polar regions (Bagshaw et al., 2013; Colombo et al., 2019; Fountain et al., 2004;
MacDonell and Fitzsimons, 2008).
In this study, we investigated the community structure and the biomass of photoautotrophs
and consumers, the isotopic composition of consumers and OM in cryoconite, and the general
elemental composition of cryoconite on the Forni Glacier, one of the most extensively studied
glacier in the Alps (e.g. Azzoni et al., 2016; Citterio et al., 2007; Franzetti et al., 2017; Pittino
et al., 2018; Senese et al., 2012, 2020; Zawierucha et al., 2019a, 2021, 2022). We tested three
main hypotheses: (i) seasonal changes in the community structure and biomass of
photoautotrophs on the Forni Glacier are reflected in the community structure and biomass of
consumers; (ii) seasonal changes in the general elemental composition of cryoconite are directly
connected with the community structure and biomass of photoautotrophs and consumers; (iii)
carbon and nitrogen stable isotopic composition of OM and consumers in cryoconite changes
during the ablation season and mirrors seasonal changes in the cryoconite community structure.
**2. Material and methods**
**2.1 Study site and sampling**
The Forni Glacier (Fig. 1) is a valley type mountain glacier located in the Ortles–Cevedale
group (Stelvio National Park, Central Italian Alps). As one of the largest glaciers in Italy with
an area of about 10.83 $km^2$, the Forni Glacier is diminishing rapidly every year. The elevation
of Forni Glacier ranges between 2600 and 3670 m a. s. l. (Senese et al., 2018). Cryoconite holes
are located on the tongue of the glacier. The lower part of the tongue is characterized by a mild
slope with small boulders and coarse grain debris on the glacier surface. The upper part of the
tongue is flat with visible patches of scattered cryoconite.



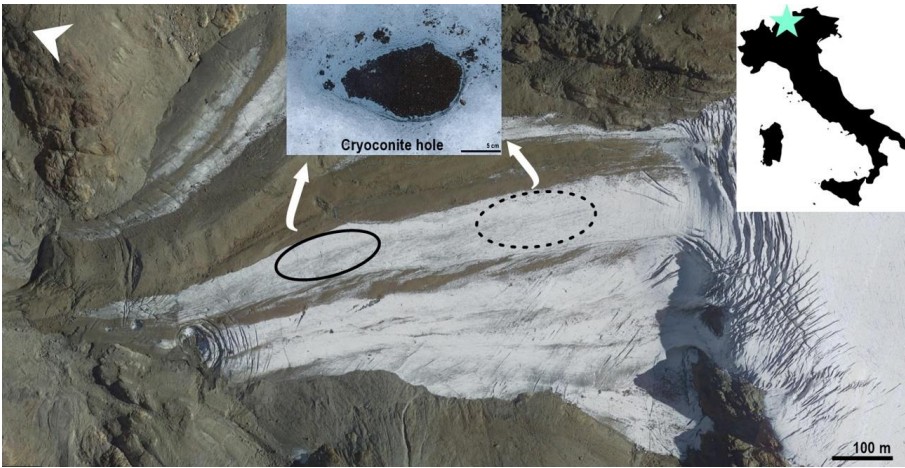


**Figure 1.** Location of sampling sites at the Forni Glacier. Solid oval indicates lower sampling sites, dotted oval indicates upper sampling sites. Source: Google Earth, version 9.172.0.0 - WebAssembly with threads (Forni Glacier 24.09.2021). © Google Earth.

Cryoconite samples were collected from the lower (approx. 2650 m a. s. l.) and the upper part (approx. 2700 m a. s. l.) of the ablation area (the area where the ice mass loss exceeds its increase) below the seracs that connect the ablation and the accumulation area (the area where the ice accrual exceeds its decrease) of the glacier (Senese et al., 2012). Horizontal distance between upper and lower sampling site was 250−300 metres. Cryoconite was collected from the bottom of cryoconite holes in five sampling campaigns during the 2019 ablation season (July 4th and 26th, August 15th and 30th, and September 19th). Samples were collected with an aseptic stainless spoon and transferred into 50 mL plastic test tubes. During each sampling campaign, cryoconite was collected from at least 5 holes to create one pooled sample at each part of the ablation zone. After collection, cryoconite was frozen and transported to a laboratory at the Adam Mickiewicz University, Poznań (Poland). Thereafter, material from each pooled sample was well mixed and split for subsequent analyses (see details below).

137

138



## 2.2 Identification and quantification of photoautotrophs

Morphological observations were conducted using a Nikon Eclipse TE2000-S digital microscope equipped with a Nikon DS-Fi1 camera under the magnification 100×. Morphometric analysis of species was conducted using NIS image analysis software. Identification of cyanobacteria, diatoms (only specimens with well-preserved, visible chloroplasts were considered for analyses), and eukaryotic green algae followed Hindak (1996), Coesel and Meesters (2007), John and Rindi (2015), Komárek and Anagnostidis (2005), and Krammer and Lange-Bertalot (1986, 1991a, 1991b). The taxonomy and nomenclature of cyanobacteria, diatoms, and eukaryotic green algae were confirmed based on Algaebase (https://www.algaebase.org/).

For quantitative analyses, each sample was analysed in 10 repetitions. For each repetition, 200 µL of analysed sample water was placed on a glass slide under a coverslip and the number of photoautotrophs were counted. Cells of photoautotrophs were counted in strips and the mean value of strips on a slide was 20. The "calculation units", namely individual cells or 100 µm filaments, were counted. Fifty specimens from each species were measured, the mean cell size or filaments were calculated with the reference to their similarity to geometric shapes according to Hutorowicz (2006). The resulting data (i.e., mean volume and number of cells in given volume) were used to calculate the biomass of individual taxa (Hutorowicz, 2006). The algal biomass is calculated by assuming that the algal cell density is 1.0 g/cm$^3$, then the algal biomass is equal to its volume.

## 2.3 Identification and quantification of top consumers

In the laboratory, 6 mL of cryoconite were analysed after slow melting in the fridge at 3 °C to avoid a heat shock to consumers. Tardigrades were identified using the original description of Mihelcic (1959), the redescription in Dastych et al. (2003), and previous description of microinvertebrates in cryoconite on the Forni Glacier (Zawierucha et al., 2019a). For the



extraction of consumers, cryoconite was placed into a Petri dish (ø 8.5 cm) and scanned for
microfauna using a stereomicroscope (Olympus BZ51). On the bottom of each Petri dish,
parallel thin lines were drawn with a black marker every 5 mm to increase the precision of
scanning (5 mm corresponds to a visible image at 30× magnification). All tardigrades were
extracted with small shovels and counted. Petri dishes were placed on the ice pad to provide
cooling for glacier invertebrates during the extraction. The density of animals was calculated
per 1 cm$^3$ and per 1 g of dry cryoconite.
**2.4 Biomass of tardigrades**
For the calculation of biomass of tardigrades, body length and width of individuals were
manually measured on photographs taken by the Quick PHOTO Camera 3.0 software (Promicra,
Prague, Czech Republic) under an Olympus BX53. Animals not suitable for measurements
(broken, bended) were not measured. Mass (wet mass (WM)) of each specimen was calculated
based on the formula of Hallas and Yates (1972): if body length (L) and width (D) were 4:1; WM
$= L^3 \times 0.051 \times 10^{-6}$, or 5:1; WM $= L^3 \times 0.033 \times 10^{-6}$.
**2.5 Organic matter in cryoconite**
The amount of organic matter in cryoconite was measured as a percentage of weight loss
through combustion (i.e., loss on ignition, LOI) at 550 °C for 3 h following drying at 50 °C for
24 h (Wang et al., 2011). The method was previously used in studies on organic matter content
in cryoconite worldwide (Rozwalak et al., 2022).
**2.6 General elemental composition of cryoconite**
*2.6.1   Procedure*
For the general elemental composition of cryoconite, combusted samples (without organic
matter) of cryoconite were used. The elements analysed were Al, As, B, Ba, Be, Bi, Ca, Cd, Ce,





Co, Cr, Cu, Dy, Er, Eu, Fe, Ga, Gd, Ge, Hf, Hg, Ho, K, La, Li, Lu, Mg, Mn, Mo, Na, Nb, Nd,
Ni, Os, P, Pb, Pr, Rb, Re, Rh, Ru, Sb, Sc, Se, Si, Sm, Sn, Sr, Ta, Tb, Te, Th, Ti, Tl, Tm, V, W,
Y, Yb, Zn, and Zr. Before starting the analyses, samples were dried at $+35 \pm 2$ ºC in an electric
oven (Thermocenter, Salvislab, Switzerland). Then, 200 mg (with the accuracy $\pm 1$ mg) of each
sample were extracted in closed Teflon containers with 5 mL of 65% nitric acid (Sigma-
Aldrich, USA) using a Mars 6 (Mars 6 Xpress, CEM USA) microwave digestion system.
Thereafter, samples were filtered and refilled to a total volume of 15 mL with Milli-Q water
(Direct-Q system, Millipore, Germany). Just before the analysis, each sample was diluted 20
times with Milli-Q water.
*2.6.2 Instrumentation*
The concentration of elements was determined by a PlasmaQuant MS Q (AnalytikJena,
Germany) inductively coupled plasma mass spectrometry. The instrumental conditions were:
plasma gas flow 9.0 L $\text{min}^{-1}$, auxiliary gas flow 1.5 L $\text{min}^{-1}$, nebulizer gas flow 1.05 L $\text{min}^{-1}$,
Radio Frequency (RF) power 1.35 kW, signal has been measured in 5 replicates (20 scans each).
The mass interferences were reduced using the integrated Collision Reaction Cell (iCRC)
working sequentially in three modes: without gas addition, with hydrogen as reaction gas and
with helium as collision gas.
*2.6.3 Analytical method validation*
The uncertainty for the total analytical procedure was below 20 %. Expanded uncertainty with
coverage factor of k = 2 (approximate 95% confidence) was calculated for all analytical steps
including sample preparation and instrumental analysis. The detection limits were calculated as
the concentration corresponding to the signal equal to three times the standard deviation of the
blank signal in the level of 0.001 mg $\text{kg}^{-1}$ of dry weight (DW). The traceability was checked by
the analysis of Standard Reference Materials (SRMs) NCS DC73349 (bush branches and


leaves, NCS Testing Technology, China), IAEA-405 (estuarine sediments, International
Atomic Energy Agency, IAEA, Austria), SRM 2709a (San Joaquin soil, National Institute of
Standards and Technology, USA); BCR-667 (estuarine sediments, Institute for Reference
Materials and Measurements, Belgium). Following the quality control requirements, the
analysis was considered valid when the results found for CRMs (certified reference material)
presented recovery were between 80% and 120%. Additionally, the standard addition method
was provided for elements with not certified values.
**2.7 Analyses of carbon and nitrogen stable isotopes**
*2.7.1  Preparation of tardigrades*
For the analyses of tardigrades, cryoconite was melted (Sect. 2.3) and tardigrades were collected
using a glass Pasteur pipette according to Novotná Jaroměřská et al. (2021). Thereafter, samples
were stored at −20 °C until further processing started. After all samples of tardigrades were
prepared, they were melted, and each individual was cleaned at least twice in a drop of distilled
water under a light microscope (Leica DM750) from superficial mineral and organic particles.
Thereafter, all tardigrades were transferred into pre-weighted tin capsules (Elemental
Microanalysis, 8 × 5 mm, D1013) using a glass Pasteur pipette. Afterwards, all samples were
stored overnight at −20 °C and at −80 °C for 1 h before the lyophilization started. The duration
of lyophilization was 2 h. Thereafter, samples were weighted (Mettler Toledo Excellence Plus
XP6; linearity = 0.0004 mg) and tin capsules were closed, wrapped, and analysed for stable
nitrogen ($\delta^{15}$N) and carbon ($\delta^{13}$C) isotopic composition. To avoid carbon contamination, all
work was conducted using nitrile gloves. Besides tardigrades, we found a few individuals of
rotifers in cryoconite. Nevertheless, their occurrence in our samples was very low (few or no
specimens among tens or hundreds of tardigrades). Therefore, rotifers were not further
analysed.
*2.7.2  Preparation of cryoconite*



For analyses of cryoconite, all animals were removed using glass Pasteur pipettes and samples
were frozen at −20 °C before further processing started. Thereafter, material was melted and
homogenized using an agate pestle and mortar and dried on a Petri dish at 45 °C for 14 h. To
avoid any contamination between samples, we partially covered all Petri dishes by an
aluminium film during the drying.
For analyses of $\delta^{15}N$ in organic matter, the dry cryoconite was transferred without any other
preparation to pre-weighted tin capsules (Costech, 9 × 5 mm, product code 41077) and
weighted. All samples were prepared in 3 replicates with an average weight ~ 29.93 mg of dry
material. Before analyses, samples were stored in a desiccator for 10 d.
For analyses of $\delta^{13}C$ in OM, ~ 0.73 mg of dry material was transferred into pre-weighted silver
capsules (Elemental Microanalysis, 8 × 5 mm, D2008) and carbonates were dissolved using
10% HCl moistened with $diH_2O$. The acid was pipetted into capsules following additions of 15,
15, 20, 50, 100 μL with drying after each addition equal or up to 50 μL according to Brodie et
al. (2011) with the modification after Vindušková et al. (2019). After the last acid addition,
samples were left drying at 50 °C for 19 h. After drying, silver capsules were inserted into tin
capsules and put into a desiccator for 10 d.
*2.7.3   Stable isotopic analyses*
Analyses of $\delta^{13}C$ and $\delta^{15}N$ in all samples were performed using a Flash 2000 elemental analyser
(ThermoFisher Scientific, Bremen, Germany) as described in Novotná Jaroměřská et al. (2021).
Released gasses ($NO_x$, $CO_2$) separated in a GC (gas chromatography) column were transferred
to an isotope-ratio mass spectrometer Delta V Advantage (ThermoFisher Scientific, Germany)
through a capillary by Continuous Flow IV system (ThermoFisher Scientific, Germany). The
stable isotope values were expressed in standard delta notation (δ) with samples measured
relative to Pee Dee Belemnite for carbon isotopes and atmospheric $N_2$ for nitrogen isotopes and
normalized to a regression curve based on international standards IAEA-CH-6, IAEA-CH-3,





IAEA 600 (IAEA, Vienna) for carbon and IAEA-N-2, IAEA-N-1, IAEA-NO-3 (IAEA, Vienna)
for nitrogen. The regression curve of the total gas for analyses of cryoconite was based on the
international standard Soil Standard Clay OAS (Elemental Microanalysis, UK). Analytical
precision as a long reproducibility for standards was within $\pm$ 0.03 ‰ for $\delta^{13}$C and $\pm$ 0.02 ‰
for $\delta^{15}$N.
The $\delta^{13}$C and $\delta^{15}$N of OM in cryoconite were used as a reference to the isotopic
composition of potential food source for the tardigrades. The $\delta^{13}$C of OM in cryoconite was
used in all statistical analyses.
**2.8 Statistical analyses**
The variations in the biomass of photoautotrophs along the season were investigated by linear
models that included sampling date, elevation (lower or upper part of the ablation zone,
dichotomous factor), and their interaction. Biomass values were log-transformed before
analyses to improve the model fit and statistical significance was assessed by a permutation
approach to account for small deviations from model assumptions. Similarly, the variation of
tardigrade biomass was related to the biomass of photoautotrophs by a linear model that also
included the elevation and their interaction as predictors.
The relative biomass of photoautotrophic groups (cyanobacteria, diatoms, and eukaryotic
green algae) was investigated with a redundancy analysis (RDA) on Hellinger-transformed
relative biomass. Stable isotopic values and elemental composition of cryoconite were analysed
using (RDA). Stable isotopic values were also compared between parts of the ablation zones
by univariate statistical tests (t-tests) whose significance was assessed with a permutation
approach because data slightly deviated from the assumptions of parametric tests.
**3. Results**
**3.1 The community structure and biomass of consumers and photoautotrophs**



Dominant invertebrates found in cryoconite were tardigrades, represented by a single species,
*Cryobiotus klebelsbergi*. Among hundreds of tardigrades, only a few bdelloid rotifers were
detected. The most abundant families of: (i) cyanobacteria were Oscillatoriaceae and
Leptolyngbyaceae, (ii) diatoms were Stephanodiscaceae, Aulacoseiraceae, and Bacillariaceae,
and (iii) eukaryotic green algae were Mesotaeniaceae, Chlorellaceae, and Oocystaceae (Table
1, Table S1).

| Sample | | 4.7 | | 26.7 | | 15.8 | | 30.8 | | 19.9 | |
|---|---|---|---|---|---|---|---|---|---|---|---|
| Elevation | | Lower | Upper | Lower | Upper | Lower | Upper | Lower | Upper | Lower | Upper |
| **Cyanobacteria** | *Chrococcus* sp. | | | | ■ | | | | | | |
| | *Phormidium* sp. | ■ | ■ | ■ | ■ | ■ | ■ | ■ | | ■ | ■ |
| | *Leptolyngbya* sp. 1 | ■ | ■ | ■ | ■ | ■ | ■ | ■ | | ■ | ■ |
| | *Leptolyngbya* sp. 2 | ■ | | | | | | | | | |
| | *Pseudanabaena* sp. | | | | | | ■ | | | | |
| **Eukaryotic green algae** | *Cylindrocystis brebbisoni* De Bary f. *cryophila* Kol | ■ | ■ | ■ | ■ | ■ | ■ | ■ | ■ | ■ | ■ |
| | *Chlorella* sp. | | ■ | | | | ■ | | ■ | | ■ |
| | *Mesotaenium* sp. | | | | | | ■ | | | | |
| | *Trochiscia granulata* Hansg. | ■ | | | | ■ | | | | | ■ |
| | *Trochiscia* sp. | | ■ | ■ | | | ■ | | ■ | | |
| | coccoid green algae | | ■ | | | | | | | | |
| **Diatoms** | *Fragilaria* sp. | | | | | ■ | ■ | | | | |
| | *Nitzchia* sp. 1 | ■ | ■ | ■ | ■ | | | ■ | ■ | ■ | |
| | *Nitzchia* sp. 2 | | | | ■ | ■ | ■ | | | | |
| | *Nitzchia* sp. 3 | | | | ■ | | | | | | |
| | *Cyclotella* sp. | ■ | ■ | ■ | | | | | | | |
| | *Pinnularia* sp. 2 | | | | ■ | | | | | | |
| | *Pinnularia* sp. 1 | | | ■ | | | | | ■ | | ■ |
| | *Eunotia* sp. | | | ■ | | | | | | | |
| | *Achnanthes* sp. | | | | | | ■ | | | | |
| | *Diatoma* sp. | | | | ■ | | | | | | |
| | *Suriella* sp. | | | | ■ | | | | | | |
| | *Aulacoseira granulata* Simonsen | | ■ | ■ | ■ | ■ | ■ | | | ■ | ■ |
| | Unidentified 1 | | | | | | ■ | | | | |
| | Unidentified 2 | | ■ | | | | | | | | |


**Table 1.** Presence (dark colour) and absence (light colour) of specific taxa of cyanobacteria,
diatoms, and eukaryotic green algae in samples from the upper (U) and lower (L) part of the
ablation zone at the alpine glacier Forni over the ablation season 2019.

Visual inspection of the data showed that at the beginning of the ablation season,
tardigrade biomass was higher in the lower part of the ablation zone than in the upper part.



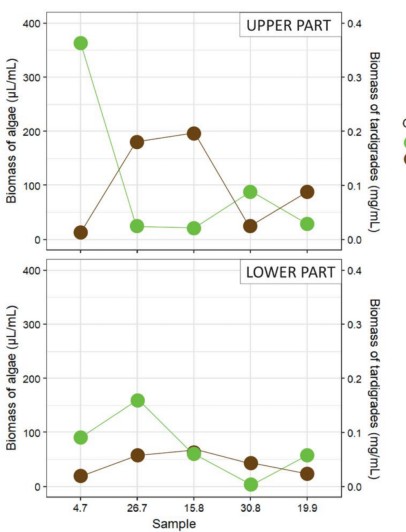

**Figure 2.** Biomass of eukaryotic green algae (μL/mL) and tardigrades (mg/mL) in the upper and lower part of the Forni Glacier tongue during the ablation season 2019.

From the first timepoint, photoautotrophic biomass in the upper part decreased while that in the lower part increased (Fig. 2).

The biomass of tardigrades in the upper part was significantly related to the

biomass of eukaryotic green algae while it did not produce any seasonal trend in the lower part
(Table 2, Fig. 3).
**Table 2.** Linear model of tardigrade biomass on chlorophyta biomass, sampling area and their
interaction. *P* values were assessed by a randomization method.

| Effect | Coef. | Adjusted SE | t | *p* |
|---|---|---|---|---|
| Intercept | 5.403 | 2.582 | 2.093 | 0.081 |
| log(Chlorophyta biomass) | −0.109 | 0.239 | −0.457 | 0.601 |
| Upper sampling area | 12.602 | 4.043 | 3.117 | 0.013 |
| log(Chlorophyta biomass) * upper sampling area | −1.096 | 0.371 | −2.951 | 0.018 |


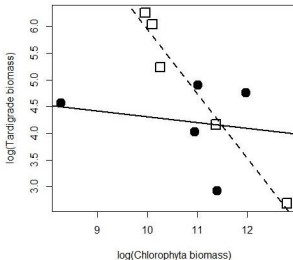

**Figure 3.** Relationship between log-transformed biomass of tardigrades and eukaryotic green algae in the lower (dots and solid line) and upper (squares and dashed line) part of the Forni Glacier ablation zone during the ablation season 2019.

Also, no significant trend in the biomass of tardigrades was found in relation to the biomass of cyanobacteria and diatoms at both parts of the ablation zones (sampling areas) over the

season ($|t_6| \leq 0.543$, *p* value $\geq 0.611$). On average, the total biomass of tardigrades in the upper
sampling area did not differ from that of the lower one ($t_8 = 0.854$, *p* value = 0.424).
The relative biomass of photoautotrophs varied between the upper and lower part of the
ablation zone (Fig. 4).



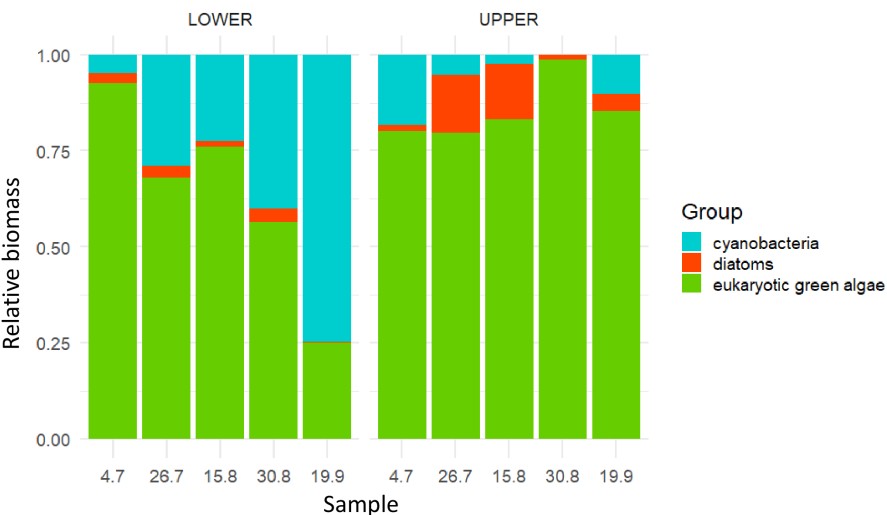

**Figure 4.** Relative biomass of photoautotrophs (cyanobacteria, diatoms, and eukaryotic green algae) in the upper and lower part of the Forni Glacier tongue during the ablation season 2019.

RDA analysis of Hellinger transformed biomass of cyanobacteria, diatoms, and eukaryotic green algae revealed that the community structure of photoautotrophs changed during the season with different patterns at each part of the ablation zone (sampling date by area interaction effect: $F_{1,6} = 6.533$, $p$ value $= 0.030$ adjusted $R^2$ of $= 0.622$, Fig. 5). In the lower part of the ablation zone, the relative biomass of cyanobacteria significantly increased during the season ($|t_6| \leq 4.735$, $P_{FDR} \geq 0.012$), the relative biomass of eukaryotic green algae decreased ($|t_6| \leq -4.642$, $P_{FDR} \geq 0.012$), and diatoms were stable ($|t_6| \leq -0.238$, $P_{FDR} \geq 0.832$). In the upper part, no taxon showed any significant trend in the relative biomass ($|t_6| \leq 0.902$, $P_{FDR} \geq 0.764$).

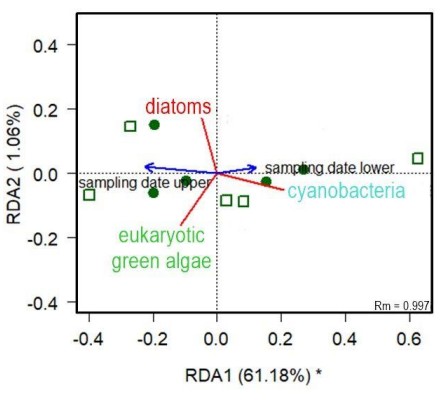

**Figure 5.** RDA correlation triplot of photoautotrophs in the upper (open squares) and the lower (full dots) part of the ablation zone. Blue arrows represent constraining covariates. $r_M$ is the Mantel correlation coefficient between distance among samples and distance among the point representing them in the plot. Values close to one indicate that the plot accurately represents reciprocal distance among samples.

349

## 3.2 Elements and organic matter in cryoconite

We observed no significant variation in OM content over the ablation season either at the upper

or the lower part of the ablation zone ($F_{3,6} = 1.238$, *p* value = 0.375; Fig. 6).

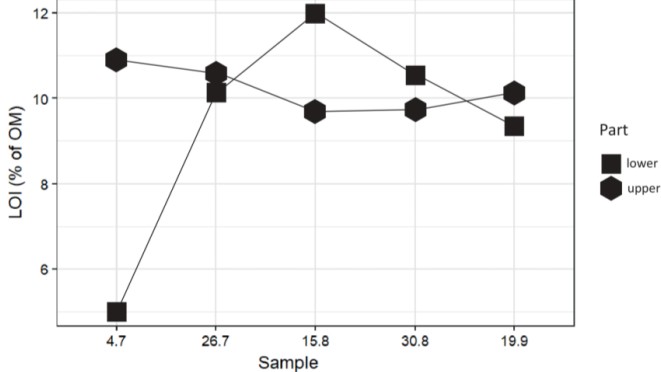

353

**Figure 6.** Percent organic matter content (LOI) in both parts of the ablation zone along the season.

356

Regarding the general elemental composition, only elements with more than 1000 µg/kg

(Ca, K, P, Si, Al, Mg) were considered. RDA models on standardized elemental abundance

showed that the elemental composition did not vary significantly according to sampling date,

part of the ablation zone, or their interaction ($F_{3,6} = 0.305$, *p* value = 0.937). The summary of

data is provided in Table 3 and Figure S1.




**Table 3.** The data on biomass of photoautotrophs and consumers, stable isotopic composition of cryoconite and consumers, organic matter content (LOI) in cryoconite, and the general elemental composition (<1000 µg/kg) of cryoconite on the alpine glacier Forni over the ablation season 2019.

| Date/Part | | 4. 7. | | 26. 7. | | 15. 8. | | 30. 8. | | 19. 9. | |
|---|---|---|---|---|---|---|---|---|---|---|---|
| | Group | Lower | Upper | Lower | Upper | Lower | Upper | Lower | Upper | Lower | Upper |
| Biomass mm³/mL | eukaryotic green algae | 89.05 | 363.46 | 160.10 | 24.24 | 60.97 | 21.14 | 3.91 | 87.31 | 57.40 | 28.49 |
| | cyanobacteria | 4.57 | 83.19 | 68.21 | 1.63 | 18.00 | 0.64 | 2.77 | 0 | 172.17 | 3.43 |
| | diatoms | 2.5 | 7.43 | 7.37 | 4.58 | 1.10 | 3.63 | 0.24 | 1.18 | 0.92 | 1.46 |
| µg/mL | tardigrades | 18.66 | 13.88 | 57.29 | 179.98 | 63.28 | 195.41 | 42.90 | 24.06 | 21.86 | 88.73 |
| LOI (%) | cryoconite | 5.01 | 10.88 | 10.15 | 10.57 | 11.97 | 9.67 | 10.54 | 9.74 | 9.34 | 10.16 |
| Stable isotopes (‰) | $\delta^{13}C$ tardigrades | −25.62 | −23.52 | −25.89 | −27.15 | −26.16 | −27.23 | x | −27.36 | −23.96 | −26.98 |
| | $\delta^{15}N$ tardigrades | x | x | −6.32 | −7.22 | −7.14 | −7.01 | x | x | x | −7.34 |
| | $\delta^{13}C$ cryoconite | −22.05 | −21.29 | −21.4 | −23.31 | −21.42 | −23.37 | −20.57 | −23.46 | −20.18 | −22.9 |
| | | −22.07 | −21.25 | −21.41 | −23.29 | −21.70 | −23.37 | −20.5 | −23.35 | −20.12 | −22.91 |
| | | −22.18 | −21.2 | −21.41 | −23.29 | −21.73 | −23.31 | −20.63 | −22.3 | −20.18 | −22.93 |
| | | x | x | x | x | x | x | x | x | x | −20.29 | x |
| | $\delta^{15}N$ cryoconite | −4.86 | −5.18 | −5.38 | −4.68 | −4.96 | −4.65 | −5.13 | −4.66 | −4.91 | −5.52 |
| | | −4.78 | −5.15 | −5.37 | −4.73 | −4.93 | −4.67 | −5.16 | −4.61 | −4.94 | −5.53 |
| | | −4.86 | −5.29 | −5.32 | −5.21 | −4.94 | −4.61 | −5.21 | −4.64 | −4.77 | x |
| Elements (mg/kg) | Ca | 2262.3737 | 2212.93 | 3011.6446 | 3318.53 | 3131.6666 | 3127.72 | 3010.48 | 2926.14 | 2580.36 | 4167.04 |
| | K | 938.1614 | 1824.23 | 1836.3847 | 2294.27 | 2489.8232 | 1854.77 | 2333.05 | 1975.69 | 2051.71 | 2486.16 |
| | P | 1475.1355 | 4619.07 | 3477.0035 | 4435.44 | 5324.6787 | 3935.42 | 4375.48 | 1298.59 | 3151.21 | 4289.11 |
| | Si | 1867.8974 | 3970.48 | 4848.4904 | 5700.19 | 6121.6709 | 4190.48 | 6865.62 | 5159.58 | 5059.43 | 7194.23 |
| | Al | 3186.3802 | 8053.97 | 7904.6442 | 11113.9 | 12475.8152 | 8990.04 | 11146.5 | 9121.65 | 8580.33 | 8981.54 |
| | Mg | 978.7658 | 2100.21 | 2158.952 | 2880.08 | 3139.3901 | 2462.2 | 3106.72 | 2375.31 | 2337.62 | 2311.53 |

### 3.3 Stable isotopic composition of organic matter and consumers

An RDA (Fig. 7a) revealed that $\delta^{13}C$ and $\delta^{15}N$ of tardigrades significantly differ from isotopic values of cryoconite ($F_{1,15}$ = 64.755, *p* value = 0.001, adjusted $R^2$ = 0.820). Tardigrades were depleted in both $\delta^{13}C$ and $\delta^{15}N$ ($t_{13} \leq$ -10.968, *p* value < 0.001) compared to cryoconite. In addition, cryoconite and tardigrades appeared to have similarities in fluctuations of their $\delta^{13}C$ values. Further analyses on stable isotopic data of tardigrades were not feasible due to the low amount of data for $\delta^{15}N$ of tardigrades (n ≤ 9), caused by a low number of specimens found in samples from some sampling campaigns.

A second RDA (Fig. 7b) with parts of the ablation zone, day-of-year, and their interaction showed that isotopic values of cryoconite changed along the melting season differently in the lower and in the upper sampling area (interaction effect: $F_{1,6}$ = 7.786, *p* value = 0.032, adjusted $R^2$ of the model = 0.693). A linear model calculated for $\delta^{13}C$ revealed that $\delta^{13}C$ values were on average higher in the lower than in the upper part of the ablation zone (coef. ± SE: 1.592 ±



0.389, $t_6 = 4.095$, $p$ value = 0.010) and changed during the season according to divergent trends
in both parts of the ablation zone (parts of the ablation zone by day-of-year interaction, $F_{1,6} =$
8.238, $p$ value = 0.035; Fig. 7c). In particular, the $\delta^{13}C$ values increased (were enriched in $^{13}C$)
over the ablation season in the lower part of the ablation zone (coef. ± SE: 0.024 ± 0.010, $t_6 =$
2.363, $p$ value = 0.045) while no relationship was observed for the upper part (coef. ± SE: -
0.017 ± 0.010, $t_6 = -1.696$, $p$ value = 0.153), even when the first and most enriched value (Fig.
7c) was removed ($t_5 = 1.196$, $p$ value = 0.275). No trend for $\delta^{15}N$ of OM in cryoconite and
consumers was observed ($F_{3,6} = 3.150$, $p$ value = 0.108).

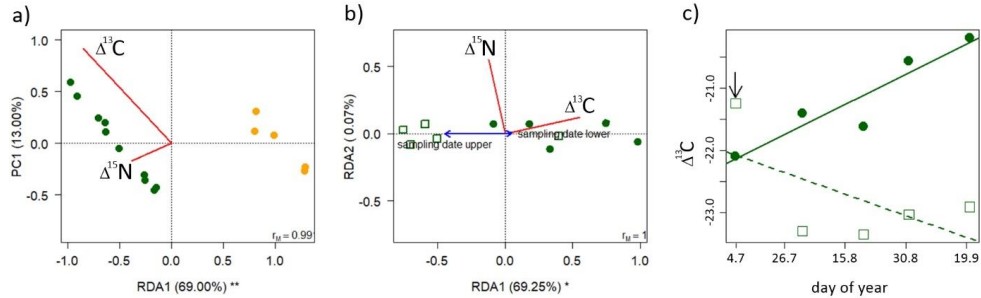


**Figure 7.** RDA correlation triplot of **a)** $\delta^{13}C$ and $\delta^{15}N$ values in cryoconite (green) and
tardigrades (orange); **b)** $\delta^{13}C$ and $\delta^{15}N$ values of cryoconite samples collected in the upper (open
squares) and lower (full dots) part of the ablation zone. Blue arrows represent constraining
covariates, $r_M$ is the Mantel correlation coefficient between distance among samples and
distance among the point representing them in the plot. Values close to one indicate that the
plot accurately represents reciprocal distance among samples; **c)** Scatterplot of $\delta^{13}C$ values in
cryoconite in the upper (open squares) and lower (full dots) part of the ablation zone. The black
arrow indicates an influential point whose removal did not change the results.


**4. Discussion**
**4.1 Photoautotrophs, elemental composition, and organic matter content**
In our study, the upper part of the ablation zone at the Forni Glacier was dominated by
eukaryotic green algae, whereas, in the lower part, the community structure changed from the
dominance of eukaryotic green algae to a dominance of cyanobacteria during the season.



Dominant eukaryotic green algae found within both parts of the ablation zone belonged to
families Mesotaeniaceae and Chlorellaceae, which are common in cryospheric habitats (Di
Bella et al., 2007; Takeuchi et al., 2001b). As discussed by Buda et al. (2020) and Vonnahme
et al. (2016), the fast growth of algae might influence their adaptation to the dynamic conditions
of cryoconite holes on valley glaciers in polar and mountain regions and lead to an increase in
biomass. In Greenland, Svalbard, and the Alps, the temporal or spatial differences in
communities of glacier cyanobacteria and eukaryotic green algae are related to meltwater
availability and physicochemical features of the environment in cryoconite holes (Di Mauro
2020; Stibal et al., 2006; Uetake et al., 2010; Vonnahme et al., 2016). For instance, Stibal et al.
(2012b) observed an increasing abundance of cyanobacteria with increasing nutrient content
and increasing distance from the glacier margin in Greenland. An increase in the proportion of
cyanobacteria at higher elevation of Greenland glaciers was also observed by Uetake et al.

416   (2010).

We did not observe a large variability in general elemental composition between both
parts of the ablation zone in our samples, even though we expected that availability of meltwater
and activity of microorganisms may influence the weathering of mineral grains (e.g. Hoppert
et al., 2004) in holes and consequently release nutrients. However, the lower part of the glacier
tongue was covered by a higher amount of proglacial debris, which could serve as a substrate
to favour the growth of cyanobacteria (Uetake et al., 2016).
Even insignificant, seasonal fluctuations and differences in OM content in the upper and
lower part of the ablation zone could indicate that the OM content is spatially dependent and
likely related to the balance between OM produced *in situ* and OM delivered from external
sources. Assuming that the upper part of the glacier tongue is more stable compared to the lower
part, a higher OM content in the upper part follows the experimental results of Buda et al.
(2021), who showed that OM is decomposed faster in dynamic conditions representing at lower





elevations. In addition, greater hydrological connectivity and slope of ice tongue may wash up
OM in lower part.
It is likely that seasonal patterns in the community structure of photoautotrophs combine
effects of phenology of glacier photoautotrophs, biological control, and physical forces shaping
their community structure. Some photoautotrophs like cyanobacterium *Phormidium* sp. or algae
*Cylindrocystis brebbisoni* dominated along the whole season in our samples. However, for
example cyanobacterium *Pseudanabaena* sp. and *Chroococcus* sp. were present only during
single sampling campaigns, and some like *Trochiscia* sp. occurred during few sampling
campaigns with untraceable presence between them.

## 4.2 Biomass of photoautotrophs and consumers

The biomass of photoautotrophs and consumers showed different seasonal trends at both parts
of the ablation zone. At the beginning of the ablation season, biomass of all photoautotrophs
decreased with increasing biomass of consumers in the upper part of the glacier, while in the
lower part the biomass of both the consumers and all photoautotrophs increased. At the end of
the season, the biomass of photoautotrophs and consumers showed opposite patterns in both
parts of the ablation zone.
On the glacier surface, we cannot exclude physical factors controlling the distribution of
biomass of photoautotrophs and consumers. Based on observations from other Arctic glaciers
(Hodson et al., 2007; Mueller and Pollard, 2004; Zawierucha et al., 2019b), meltwater may be
an important factor in redistribution of cryoconite along the glacier surface. Thus, at the
beginning of the season, meltwater may wash the cryoconite and sediment down from the upper
part of the glacier and cause the input of cryoconite with photoautotrophs to the lower parts as
observed by Takeuchi et al. (2001b).
Nevertheless, biological control may be also crucial in cryoconite hole ecosystem
functioning (Cook et al., 2016; McIntyre, 1984). Scheffer et al. (2008) suggested that if





densities of consumers are low, algae can escape from top-down control. The observation of
glacier tardigrade *C. klebelsbergi* under laboratory conditions revealed that this species actively
feeds on a mix of *Chlorella* and *Chlorococcum* both belonging to Chlorophyta (K. Zawierucha
pers. observ.). Moreover, Zawierucha et al. (2022) showed that in the field *C. klebelsbergi* feed
on the eukaryotic green algae (Trebouxiophyceae).
We observed a negative relation between the biomass of eukaryotic green algae and the
biomass of tardigrades in the upper part of the glacier tongue. In the same part, the only
sampling date with reduced biomass of tardigrades accompanied with an increase in the
biomass of eukaryotic green algae was affected by the presence of numerous small tardigrade
juveniles (K. Zawierucha pers. observ.) likely decreasing the overall biomass of consumers and
potentially favour the growth of algae. Indeed, Vonnahme et al. (2016) suggested that
microalgae in cryoconite holes can increase their densities (cell size, formation of colonies) as
a response to the grazing pressure.
On the contrary, the biomass of photoautotrophs in the lower part of the ablation area
seemed to be affected by temporal or episodic changes more than consumers, which remained
almost stable with a slight increase in their biomass at the beginning of the season and a slight
decrease at the end. Although, Scheffer et al. (2008) suggested that if organisms are slow-
growing, they are much less affected by episodic pulses (e.g. mirroring the lower part of the
ablation zone), the biomass in the lower part, of both algae and grazers, didn't build up so fast
as in the upper part, most probably due to less of stability. However, our assumptions are based
on observation from one season only and require additional testing in the future.

**475   4.3 δ¹³C and δ¹⁵N isotopic composition**

Changes in irradiation, higher photosynthetic activity, higher growth rate or differences in the
nutrient pool could change the proportion of carbon and nitrogen forms in the cryoconite and
consequently affect the isotopic values of the biota (e.g. Beardall et al., 1982; Gu et al., 2006;





Lehmann et al., 2004; Senese et al., 2016; Schmidt et al., 2022; Yoshii et al., 1999). The more
depleted $\delta^{13}C$ and $\delta^{15}N$ of tardigrades compared to the cryoconite organic matter on the Forni
Glacier corroborates the results from Arctic cryoconite holes (Novotná Jaroměřská et al., 2021)
and microbial mats in Antarctica (Almela et al., 2019; Velázquez et al., 2017). However, since
microbial mats are different systems, organic matter in cryoconite holes on the Forni Glacier
was depleted in heavy carbon ($^{13}C$) and nitrogen ($^{15}N$) isotopes and the differences between
$\delta^{13}C$ and $\delta^{15}N$ of organic matter and consumers were higher. Based on previous models (Almela
et al., 2019; Velázquez et al., 2017), tardigrades likely fed on cyanobacteria, diatoms, and POM
(particulate organic matter) < 30 μm. Even though cryoconite from the Forni Glacier contains
consumable cyanobacteria and algae, our results do not correspond with the standard
fractionation between consumer and food (DeNiro and Epstein, 1978; Yoshii et al., 1999).

490       Since tardigrades in all samples followed fluctuations of $\delta^{13}C$ values of cryoconite, we

suggest that some components of cryoconite serve indeed as their food source (Novotná
Jaroměřská et al., 2021). Nevertheless, our results are probably highly influenced by the stable
isotopic composition of the unconsumed part of cryoconite, which increases the differences
between food and consumers.

495       Despite the significance of autochthonous production of microbes, most of the organic

matter in cryoconite holes seems to be of allochthonous origin (Stibal et al., 2008). Forni is a
relatively small glacier and the allochthonous material covers the whole ablation zone with its
inorganic part predominantly originated from surrounding rocks (Azzoni et al., 2016).
Therefore, the low $\delta^{13}C$ of OM in cryoconite from the Forni Glacier compared to cryoconite
from Antarctic glaciers with higher occurrence of photosynthetically active cyanobacteria
(Schmidt et al., 2022) or microbial mats (Almela et al., 2019; Velázquez et al., 2017) may be
influenced by the prevailing allochthonous organic matter which can lower the $\delta^{13}C$ compared
to material formed *in situ* (Musilova et al., 2015; Pautler et al., 2013; Stibal et al., 2008).





The differences in $\delta^{13}C$ of OM in cryoconite between parts of the ablation zone and the
increasing seasonal trend in $\delta^{13}C$ in the lower part of the glacier tongue can be the result of the
seasonal evolution in microbial community structure and the dominance of *in situ* microbial
production predominantly using isotopically heavy DIC (dissolved inorganic carbon) instead
of atmospheric $CO_2$ (Musilova et al., 2015; Stibal and Tranter, 2007). Communities of
eukaryotic green algae were dominated by Chlorellales and Zygnematales in both parts of the
ablation zone. Based on Beardall et al. (1982), nitrogen limitation in *Chlorella emersonii* results
in higher $\delta^{13}C$ values due to the higher accumulation of $CO_2$ and lower fractionation against
$^{13}C$ by RuBiSCO (ribulose 1,5 bisphosphate carboxylase-oxygenase). The $\delta^{15}N$ ratios were not
different in cryoconite between both parts of the ablation zone. Also, we were unable to analyse
the isotopic composition of each group of photoautotrophs separately, so we could not reveal
their contribution to the overall isotopic signal of OM in cryoconite.
**5. Conclusions**
In this study, we described spatial changes in the community structure, biomass, and stable
carbon and nitrogen isotopic composition of biota from cryoconite holes in the ablation tongue
of the alpine glacier Forni during the summer season. Since we did not observe any significant
fluctuations in the general elemental composition of cryoconite, changes in the composition
and biomass of photoautotrophs and consumers in both parts of the ablation zone indicated
phenological or ecological controls over their communities. Some photoautotrophs appeared
only during specific sampling days pointing out that rare species might be overlooked during
single sampling campaigns. Based on our data we assume that photoautotrophs in cryoconite
holes might be controlled by grazing; they may increase their biomass as a protection against
overgrazing or escape from top-down control. However, other factors such as influence of
meltwater, weathering, or the input of matter from adjacent sources cannot be overlooked and
require further investigation in studies on seasonal development of cryoconite community in



the future. Seasonal increase in $\delta^{13}$C in the lower part of the glacier tongue may suggest
potential changes in the microbial community structure, nutrient concentration, or differences
in the source of OM. We demonstrated that the recognition of the community structure of
cryoconite holes requires a broad-scale and seasonal approach since biological communities
vary in time and space on the glacier surface.
*Data availability.* All data are available upon request to TNJ and KZ.
*Author contributions.* TNJ performed research, analyzed data, and wrote the paper. RA
analyzed data and wrote the paper. DR, MP, PN, JS-K and PK performed research and analyzed
data. AF and FP performed research. LV analyzed data. TK performed research and analyzed
data. KZ conceived and designed study, performed research, analyzed data, and wrote the paper.
All authors contributed significantly to the redaction of the paper.
*Competing interests.* The authors declare that they have no conflict of interest.
*Acknowledgments.* We would like to thank Sabina Sikorska, Aleksandra Kozłowska and
Masato Ono for their help in biomass measurement and their support in preparation of
supplementary figures.
*Financial support.* This research (analyses on organic matter, biomass measurements, the
investigation of food preference and biotic control) was supported by the grant OPUS funded
by National Science Centre (no. 2018/31/B/NZ8/00198) awarded to KZ. Analyses of stable
isotopic composition were supported by a Charles University research grant GA UK (Charles
University Grant Agency), grant no. 596120 awarded to TNJ.





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
