# Peer review of "Spatio-temporal changes in cryoconite community, isotopic, and elemental composition"

_Biogeosciences, 2022_

## Referee Comment (RC1)

General comments:

An interesting and generally well written study on invertebrates (particularly tardigrades) and their relationship to photoautotrophic species in cryoconite on the glacier surface.

Specific comments:

Abstract
Line 23: "the" terrestrial cryosphere
Line 24: how many cryoconites were sampled?
Line 33: can be reduced to "Some phototrophic taxa"
Line 36: give statistic

Introduction
Line 47-52: I would actually begin with a topic sentence on glaciers, and then how quickly they are changing, to what diversity glacier host on their surfaces, and then why this matters.
Line 84-85: a little unclear, suggest rewording to: "together with how tardigrades interact with the food web".

Materials and Methods:
Line 126: define ablation zone where it is first mentioned in the intro
Line 132: "stainless steel" spoon
Line 130: how many cryoconites were sampled?

Results:
Line 286: Is there a percentage or number of rotifers you can include here for reference?
Line 287-289: Would be useful to denote which species in the table belong to which family
Line 309: change to "however, no seasonal trend in the lower part was observed" for clarity

Discussion:
Line 249: term "wash up" is a little vague- do you mean that ice marginal melt and other melt dynamics result in washing away of DOM?
Line 472: due to less stability

Figures/Tables:
Figure 1: Would be nice to see where each sample lies within the upper and lower part- would it be possible to add their locations on the map?
Table 1: to clarify, this table is showing in which samples there is at least one of the species shown?
Table 2: Unsure if the repeated line "upper sampling area" is meant to be there, and why is log (Chlorophyta biomass) present twice? Suggest showing results from modelling season as well.
Figure 5: would it be possible to include a legend with the figure denoting upper and lower zone samples?

Figure 7: Like Figure 5, I think a legend underneath each RDA would be nice, just to cut down on text in the caption